# The differential regulation of placenta trophoblast bisphosphoglycerate mutase in fetal growth restriction: preclinical study in mice and observational histological study of human placenta

Sima Stroganov[1], Talia Harris[2], Liat Fellus-Alyagor[3], Lital Ben Moyal[1], Romina Plitman Mayo[1], Ofra Golani[4], Dana Hirsch[4], Shifra Ben-Dor[4], Alexander Brandis[4], Tevie Mehlman[4], Michal Kovo[5,6], Tal Biron-Shental[5,6], Nava Dekel[1], Michal Neeman[1]*

[1]Immunology and Regenerative Biology, Weizmann Institute of Science, Rehovot, Israel; [2]Chemical Research Support Weizmann Institute of Science, Rehovot, Israel; [3]Veterinary Resources, Weizmann Institute of Science, Rehovot, Israel; [4]Life Science Core Facilities, Weizmann Institute of Science, Rehovot, Israel; [5]OBGYN, Meir Medical Center, Kfar Saba, Israel; [6]Tel Aviv University, School of Medicine, Tel Aviv, Israel

*For correspondence:
michal.neeman@Weizmann.ac.il

## Abstract

**Background:** Fetal growth restriction (FGR) is a pregnancy complication in which a newborn fails to achieve its growth potential, increasing the risk of perinatal morbidity and mortality. Chronic maternal gestational hypoxia, as well as placental insufficiency are associated with increased FGR incidence; however, the molecular mechanisms underlying FGR remain unknown.

**Methods:** Pregnant mice were subjected to acute or chronic hypoxia (12.5% $O_2$) resulting in reduced fetal weight. Placenta oxygen transport was assessed by blood oxygenation level dependent (BOLD) contrast magnetic resonance imaging (MRI). The placentae were analyzed via immunohistochemistry and in situ hybridization. Human placentae were selected from FGR and matched controls and analyzed by immunohistochemistry (IHC). Maternal and cord sera were analyzed by mass spectrometry.

**Results:** We show that murine acute and chronic gestational hypoxia recapitulates FGR phenotype and affects placental structure and morphology. Gestational hypoxia decreased labyrinth area, increased the incidence of red blood cells (RBCs) in the labyrinth while expanding the placental spiral arteries (SpA) diameter. Hypoxic placentae exhibited higher hemoglobin-oxygen affinity compared to the control. Placental abundance of Bisphosphoglycerate mutase (BPGM) was upregulated in the syncytiotrophoblast and spiral artery trophoblast cells (SpA TGCs) in the murine gestational hypoxia groups compared to the control. Hif1α levels were higher in the acute hypoxia group compared to the control. In contrast, human FGR placentae exhibited reduced BPGM levels in the syncytiotrophoblast layer compared to placentae from healthy uncomplicated pregnancies. Levels of 2,3 BPG, the product of BPGM, were lower in cord serum of human FGR placentae compared to control. Polar expression of BPGM was found in both human and mouse placentae syncytiotrophoblast, with higher expression facing the maternal circulation. Moreover, in the murine SpA TGCs expression of BPGM was concentrated exclusively in the apical cell side, in direct proximity to the maternal circulation.

**Conclusions:** This study suggests a possible involvement of placental BPGM in maternal-fetal oxygen transfer, and in the pathophysiology of FGR.

**Funding:** This work was supported by the Weizmann Krenter Foundation and the Weizmann – Ichilov (Tel Aviv Sourasky Medical Center) Collaborative Grant in Biomedical Research, by the Minerva Foundation, by the ISF KillCorona grant 3777/19.

## Editor's evaluation

This valuable study presents findings on the role of BPGM enzyme and its product 2,3-BPG in placental oxygenation during pregnancy, particularly in the context of fetal hypoxia and fetal growth restriction. The solid evidence draws on a comprehensive array of methods including tissue histology, high-resolution tissue MR imaging, and biochemical analysis in both mouse models and human subjects. The major strength of the study lies in the novel exploration of BPGM in placental oxygenation, while further investigations could expand on these findings to establish more definitive connections between the enzyme's activity and fetal health outcomes. The findings have practical implications for the subfield of developmental biology, particularly in understanding the mechanisms of fetal growth restriction. This research will predominantly interest developmental biologists and medical professionals specializing in prenatal care.

## Introduction

The placenta is a transient organ, crucial for the growth and development of the fetus during gestation (*Garnica and Chan, 1996*, *Gude et al., 2004*). The placenta provides the interface between the maternal and fetal circulation, mediating gas and metabolic exchange along with fetal waste disposal (*Burton and Fowden, 2015*). Abnormalities in placental growth, structure, and function are associated with gestational complications such as FGR (*Woods et al., 2018*, *Sun et al., 2020*), which is defined as the failure of the fetus to reach its growth potential (*Romo et al., 2008*). The clinical definition of FGR is fetal weight below the 10th percentile of predicted fetal weight for gestational age (*Lausman et al., 2013*). FGR affects approximately 10–15% of pregnancies, increasing the risk of perinatal morbidity and mortality (*Romo et al., 2008*). Long-term complications of FGR include poor postnatal development and are associated with multiple adverse health outcomes including respiratory, metabolic and cardiovascular deficits (*Sharma et al., 2016*, *von Beckerath et al., 2013*).

Multiple factors contribute to FGR, including fetal genetic aberrations or malformations, placental or umbilical cord defects, and maternal infections or diseases. Maternal anemia, smoking, high altitude residency, as well as placental and umbilical cord anomalies, are all associated with restricted placental and fetal oxygen availability (*Chew and Verma, 2022*). Interestingly, about 40% of all FGR cases are idiopathic (*Ghidini, 1996*), with no identifiable cause, which might hint on possible biological predisposition factors that contribute to FGR development by creating an hypoxic placental or embryo environment. However, the molecular mechanisms that provoke and contribute to this pregnancy complication have yet to be elucidated.

One of the key placental functions is the transfer of oxygen from the mother to the fetus (*Burton, 2009*), and inefficient oxygen transport and availability is detrimental for placental and embryonic development (*Carter, 2015*, *Zhou et al., 2013*). Late-gestation hypoxia results in utero-placental vascular adaptations, such as capillary expansion, thinning of the inter-haemal membrane and increased radial artery diameters (*Cahill et al., 2018*). Moreover, there is substantial evidence that late-gestation exposure to hypoxic environment alters placental structure and functionality (*Higgins et al., 2016*, *Tomlinson et al., 2010*). In-vitro studies on human placental samples under acute reduction of oxygen tension induced direct placental vasoconstriction (*Howard et al., 1987*). Placental oxygen transport depends on Hemoglobin (Hb), which is responsible for carrying and mediating oxygen transfer in mammalian organisms (*Mairbäurl and Weber, 2012*). BOLD contrast MR imaging is a powerful tool that utilizes hemoglobin as an endogenous reporter molecule to assess oxygen-hemoglobin affinity (*Avni et al., 2015*). Previous MR studies have shown altered placental oxygen-Hb affinity following exposure to hypoxia (*Avni et al., 2016*). However, limited information is available on how placental structure and function is altered in chronic gestational hypoxia that commences at the onset of gestation.

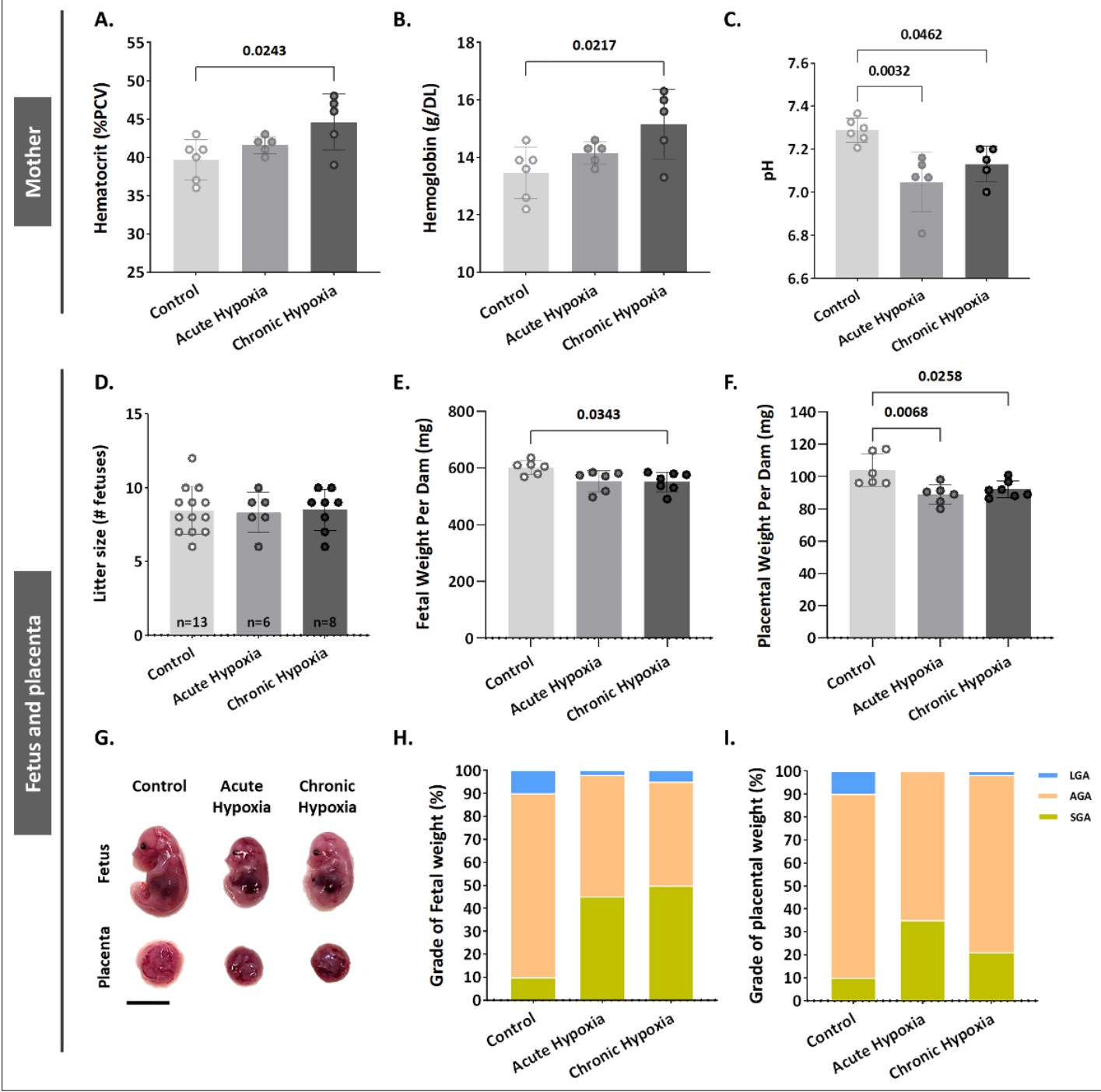

**Figure 1.** Gestational hypoxia elevates maternal hemoglobin, hematocrit and blood acidity, and recapitulates FGR phenotype in mice. (**A–B**) Graphs showing hematocrit and hemoglobin levels in maternal venous blood. (**C**) Graph shows pH levels in maternal venous blood. (**D–F**) Graphs showing litter size, fetal weight and placental weight. (**G**) Representative picture of fetuses and placentae (E16.5) from control and gestational hypoxia groups. (**H–I**) Analysis of the percentage of small for gestational age (SGA, weight less than the 10th percentile) fetuses and placentae, large for gestational age (LGA, weight greater than the 90th percentile) fetuses and placentae, and appropriate for gestational age (AGA, weight between the 10th and 90th percentiles) fetuses and placentae at E16.5. Scale bars: 1 cm. Data displayed as mean ± SD and are from 49 to 62 fetuses and placentae from 6 to 7 dams per group (8–9 conceptuses per litter used). Ordinary one-way ANOVA test was used for statistical analysis.

The most significant allosteric effectors of Hb are organic phosphates, specifically 2,3 BPG, which is produced by the BPGM enzyme in a unique side reaction of glycolysis, known as the Luebering-Rapoport pathway (*Sasaki et al., 1982*). 2,3 BPG plays a key role in delivering $O_2$ to tissues by binding to and stabilizing deoxy-hemoglobin, thus leading to the release of oxygen from the Hb unit (*Mulquiney and Kuchel, 1999*, *Steinberg et al., 2018*). During gestation, fetal hemoglobin (HbF) is the dominant form of Hb present in the fetus, comprised of α and γ subunits. During late gestation, the γ subunit is gradually replaced by the adult β subunit (*Bauer et al., 2012*). HbF has a higher affinity to oxygen compared to the adult Hb, caused by a structural difference, which leads to a weakened ability to bind 2,3 BPG (*Bauer et al., 1968*, *Frier and Perutz, 1977*, *Adachi et al., 1997*). The transfer of oxygen from maternal to fetal Hb is facilitated by the higher affinity of maternal Hb to 2,3 BPG (*Steinberg et al., 2018*). Remarkably, BPGM expression is specifically restricted to erythrocytes and the syncytiotrophoblast of the placenta, a multinucleated layer that mediates transport of oxygen and nutrients from the mother to the fetus (*Pritlove et al., 2006*). In a study that used $Igf2^{+/-}$ knockout mice as a model of FGR, BPGM expression in the placental labyrinth was lower compared to wild-type placentae (*Gu et al., 2009*). However, scarce information is available on the role of this enzyme during gestation. We report here that placental BPGM expression pattern is consistent with a role in adaptation of the placenta to gestational hypoxia, facilitating the transfer of oxygen from maternal to fetal circulation. Here, we show that gestational hypoxia augments placental BPGM expression in mice, while in human FGR placentae of unknown etiology BPGM expression is suppressed.

## Methods

### Animals

Female C57BL/6JOlaHsd mice (8–12 weeks old; Envigo, Jerusalem; n=28) were mated with C57BL/6JOlaHsd male mice (Envigo, Jerusalem; n=8). Detection of a vaginal plug the following day was considered embryonic day 0.5 (E0.5). At E0.5 or E11.5, the pregnant females were randomly allocated to control (21% $O_2$, n=15) or hypoxia group (12.5% $O_2$, acute hypoxia; n=6, chronic hypoxia; n=7). Throughout the experiments, the animals were maintained in a temperature-controlled room (22 ± 1°C) on a 12 hr:12 hr light–dark cycle. Food and water was provided ad libitum and animal well-being was monitored daily. At E16.5 the pregnant females were analyzed using high-field MRI under a respiration challenge of hyperoxia-to-hypoxia (40% $O_2$, 20% $O_2$, 10% $O_2$). After MR imaging, the animals were sacrificed via cervical dislocation for tissue collection. All experimental protocols were approved by the Institutional Animal Care and Use Committee (IACUC) of the Weizmann Institute of Science, Protocol number: 07341021–2.

### Establishment of maternal hypoxia models

We applied two models of maternal hypoxia – acute and chronic. The pregnant mice were housed in a hypoxic chamber (VelO2x, Baker Ruskinn, Sanford, Maine, USA) from E11.5 (acute hypoxia; n=6) or E0.5 (chronic hypoxia; n=7) until E16.5. On the first day in the hypoxic chamber, maternal oxygen supply was gradually reduced from 21% $O_2$ to 12.5 ± 0.2% $O_2$ by continuous infusion of a nitrogen gas. The water contained in the expired gas was trapped using silica gel beads (Merck, CAS #: 7631-86-9). A portable oxygen analyzer ($PO_2$-250, Lutron, Coopersburg, Pennsylvania, United States) was used to monitor the oxygen concentration in the chamber. Pregnant control females were housed in an identical chamber supplied with a constant 21% ± 0.2% $O_2$ concentration.

### In vivo MR imaging

MR imaging examinations were performed at a 15.2T with an MR spectrometer (BioSpec 152/11 US/R; Bruker, Karlsruhe, Germany) equipped with a gradient-coil system capable of producing pulsed gradients of 10 mT/cm in each of the three orthogonal directions. A quadrature volume coil with a 35 mm inner diameter and an homogeneous radiofrequency field of 30 mm along the axis of the magnetic field was used for both transmission and reception. Immediately prior to MR imaging, the pregnant females were anesthetized with isoflurane (3% for induction; Piramal, Mumbai, India) mixed with 2 L/min of 40% $O_2$ and 60% $N_2$ delivered into a closed induction chamber. Once anesthetized, the animals were placed in a prone position in a head holder with breathing gas mixed with isoflurane delivered through a tooth bar. Respiration rate and rectal temperature were monitored using a

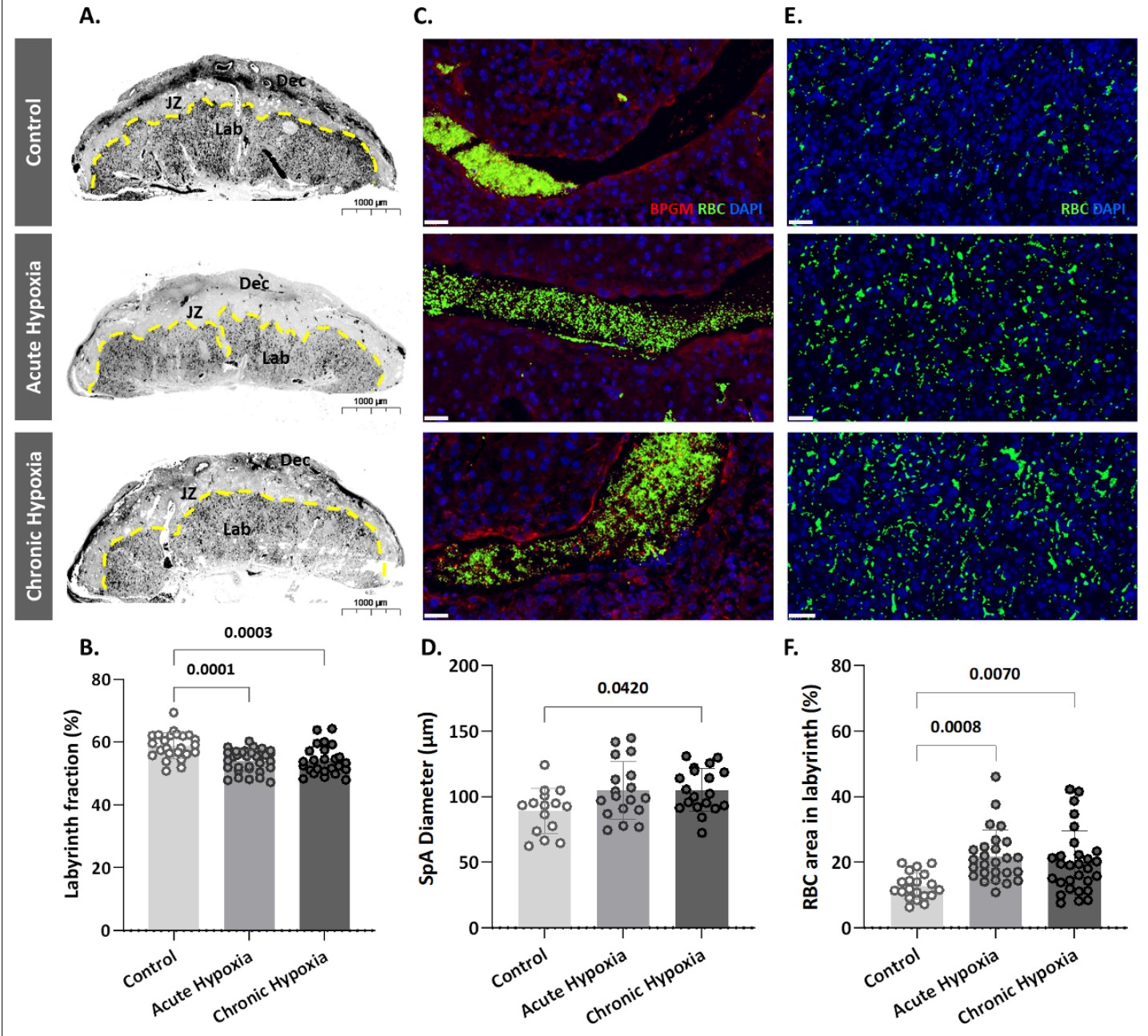

**Figure 2.** Maternal hypoxia during gestation results in enlarged spiral arteries, increased RBC levels and decreased placental labyrinth area. (**A, B**) Placentae of hypoxic chamber groups have significantly smaller labyrinth area in comparison to the control group. (**C, D**), (**E, F**) Placentae of hypoxic chamber groups display enlarged spiral arteries and increased RBC levels in the labyrinth. Scale bars: 40 µm. Data are from 3 control, 4 chronic hypoxia and 4 acute hypoxia dams, 5–7 placentae per dam and presented as mean ±SD values. Ordinary one-way ANOVA test was used for statistical analysis.

monitoring and gating system (Model 1030 S-50; SA Instruments, Stony Brook, NY). Respiration rate was maintained throughout the experimental period at approximately 20–30 breaths per minute by adjusting the isoflurane level (1–2% for maintenance). Body temperature was maintained at 30 ± 1°C (to reduce fetal movement) by adjusting the temperature of a circulating water heating blanket placed above the animal.

## MR imaging data acquisition

Anatomic data to determine optimal animal positioning was acquired by using a short Gradient Recalled Echo (GRE) sequence with imaging slices acquired in three orthogonal planes. The animals

were positioned to maximize the number of fetuses that could be viewed while still observing maternal liver. The duration of the MRI measurements at each oxygen level was approximately 20 min. After the $O_2$ concentration was reduced, a 2-min interval was given before acquiring the next set of MRI images, allowing relaxation rate (R2*) stabilization. At each oxygen phase, the nitrogen level was adjusted to maintain a constant flow of inhaled gas. To determine R2* values three Gradient Recalled Echo (GRE) acquisitions were performed with TE = 1.6 ms, 2.6ms, and 3.6ms. The parameters for these GRE measurements were as follows: 48 slices with slice thickness of 0.4 mm with 0.1 mm inter-slice gap, field of view 4.2X3.3 $cm^2$, pulse flip angle 40°, matrix size 280x220 (150 x 150 $um^2$ pixel size), 2 averages (motion averaging). Images were acquired with fat suppression and RF spoiling. The excitation pulse was 0.5ms (6400 Hz bandwidth) and the acquisition bandwidth was 200 kHz. The slice order was interleaved. The sequence was respiration triggered (per slice) with an approximate TR of 800ms.

## MR imaging data analysis

Images were reconstructed by Paravision 6.0 (Bruker, Karlsruhe, Germany). The GRE images used for calculating R2*s were interpolated in Matlab (MathWorks, Natick, Massachusetts, USA) to 75X75 $\mu m^2$ pixel size. Regions of Interest (ROIs) were manually marked with ImageJ (U. S. National Institutes of Health, Bethesda, Maryland, USA). Subsequently, using custom written scripts all ROIs and images were imported into Matlab and the R2* for each $O_2$ level was determined by fitting the changes in the median signal intensity of each ROI to a single exponential decay (*Equation 1*).

$$Int = Int_0 \cdot e^{-R_2^* \cdot TE} \tag{1}$$

## Tissue collection

### Mouse placentae samples

After MR Imaging of the animals, maternal blood was collected from the submandibular vein, followed by cervical dislocation. Maternal hematocrit, Hb and pH levels were determined using i-STAT CG8 +cartridge (Abbott, Cat. No. ABAXIS-600-9001-10, Chicago, Illinois, USA). Uterine tissues were immersed in PBS to count the number of fetuses and resorptions. Fetuses and placentae were immediately removed and weighed, following by fixation in 4% paraformaldehyde. Grade of embryonic and placental weight was classified as SGA (weight less than the 10th percentile), appropriate for gestational age (AGA, weight between the 10th and 90th percentiles) and large for gestational age (LGA, weight greater than the 90th percentile).

### Human placentae samples

The study was approved by the Meir and Wolfson Medical Center IRB Local Committee (Protocols: # 0147–20 MMC and #185–19-WOMC). Written informed consent was obtained from all participants prior to delivery. Placentae from nine healthy uncomplicated pregnancies and from seven pregnancies complicated by fetal growth restriction (FGR) were collected immediately after elective cesarean deliveries. Two biopsies were taken from each placenta, one from a peripheral and one from a central lobule. The biopsied material (~1 $cm^3$) was immediately fixed in formalin. FGR birth weight standards were based on the Dollberg curve (*Dollberg et al., 2005*).

### Human serum

Maternal and cord serum samples were collected from the enrolled patients prior to delivery, and from the umbilical cord just following delivery. The umbilical cord was wiped clean and blood was drawn from the vein. Blood samples were centrifuged (1000 × *g*, 10 min at room temperature), and serum aliquots were stored at –80 °C in dedicated tubes for analyses at the Weizmann Institute. The CDC hemolysis reference palette was used to exclude the hemolysed samples.

## Immunohistochemistry and microscopy

Fixed murine and human placentae were processed and embedded in paraffin. Representative 5 μm sections were taken from each tissue and used for IHC.

All slides were dewaxed and rehydrated in xylene and a series of ethanol washes. IHC staining involved antigen retrieval in a pressure cooker using citrate buffer (pH = 6) and blocking of non-specific binding with 20% NHS and 0.2% Triton in PBS. Slides were incubated with polyclonal rabbit

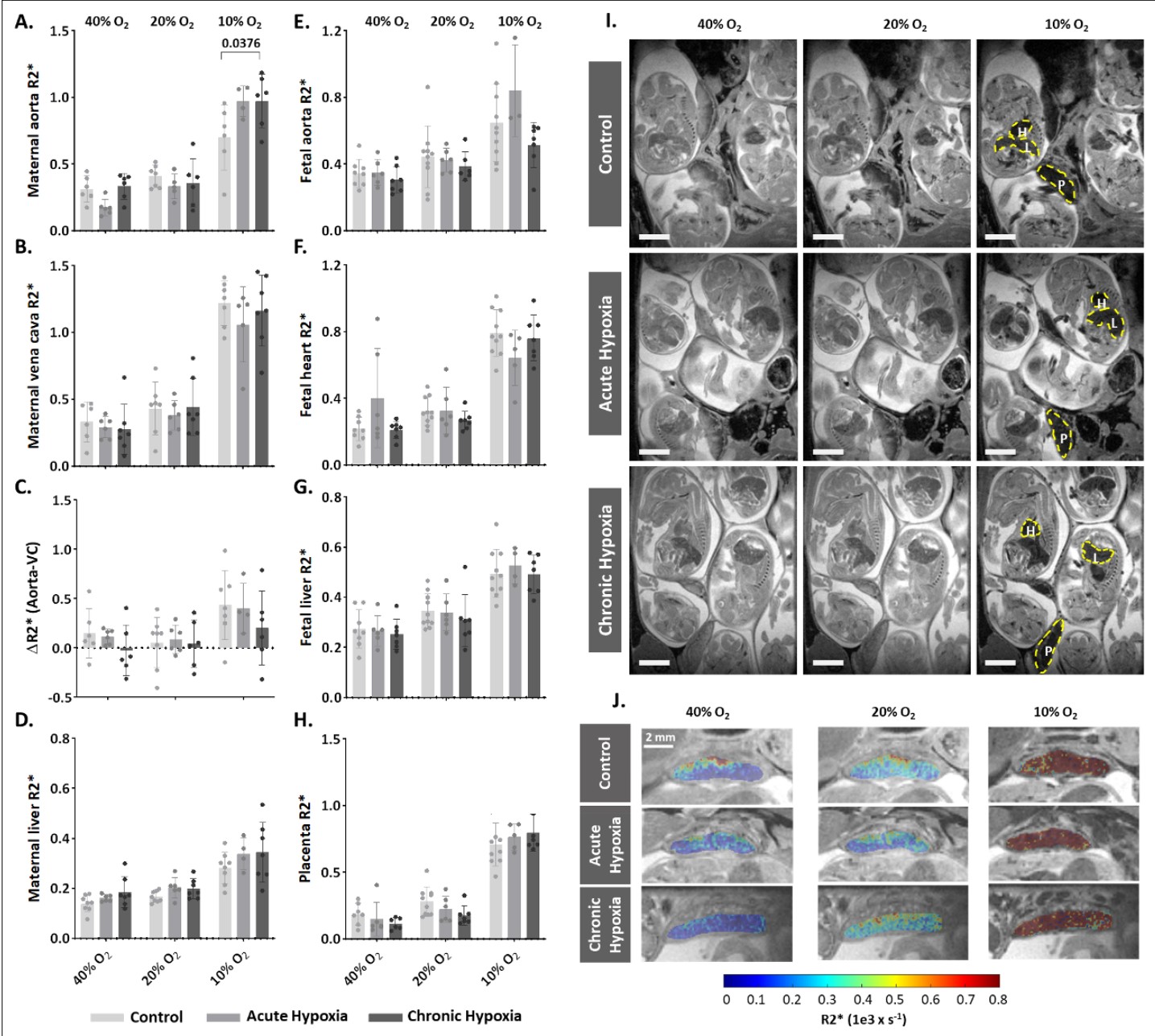

**Figure 3.** Effects of maternal hypoxia during gestation on R2* values following hyperoxia-hypoxia challenge. (**A–H**) Graphs show that hypoxic challenge results in elevation in R2* values in maternal aortas of chronic hypoxia chamber group, while no differences are observed in the respective placentae and fetuses. (**I**) Representative R2* images of control and hypoxic chamber group show several fetuses and their placenta (**P**), heart (**H**) and liver (**L**). Scale bars: 0.5 cm. (**J**) Representative R2* maps inside the placenta of control, acute hypoxia (AH) and chronic hypoxia (CH) chamber groups at E16.5 show distribution of R2* values following hyperoxia-hypoxia challenge. Data are from 8 control, 6 acute hypoxia and 7 chronic hypoxia per dams presented as mean ± SD values. R2* values of embryonic tissues and placentae are calculated as the median per mother, 5–8 embryos per each mother. Ordinary one-way ANOVA test was used for statistical analysis.

The online version of this article includes the following video and figure supplement(s) for figure 3:

**Figure supplement 1.** Effects of maternal hypoxia during gestation on R2* values following hyperoxia-hypoxia challenge.

**Figure 3—video 1.** MR imaging of mother, embryos, and placentae: Representative MRI scan videos of control dams.
https://elifesciences.org/articles/82631/figures#fig3video1

**Figure 3—video 2.** MR imaging of mother, embryos, and placentae: Representative MRI scan videos of acute hypoxia dams.
https://elifesciences.org/articles/82631/figures#fig3video2

*Figure 3 continued on next page*

Figure 3 continued

**Figure 3—video 3.** MR imaging of mother, embryos, and placentae: Representative MRI scan videos of chronic hypoxia dams.
https://elifesciences.org/articles/82631/figures#fig3video3

primary anti-BPGM antibody (1:200, Sigma-Aldrich, Cat. No. HPA016493, RRID: AB_1845414), followed by incubation with an HRP anti-Rabbit secondary antibody (1:100, Jackson ImmunoResearch Labs, Cat# 111-035-003, RRID: AB_2313567) followed by Opal 690 (1:500, Akoya Biosciences, Cat. No. FP1497001KT). Negative controls for each immunostaining were incubated with secondary antibody only.

Images were captured using Nikon Eclipse Ti2_E microscope, Yokogawa CSU W1 spinning disk, photometrics Prime 25B camera with NIS elements AR 5.11.01 64bit software.

For co-detection of BPGM with SynI, SynII, HIf1a and Hif2a, HCR IF +HCR RNA-FISH protocol for FFPE sections was employed (Molecular Instruments; *Schwarzkopf et al., 2021*) according to manufacturer instructions using an antibody for BPGM (cat num, company, 1:50, antigen retrieval with PH = 6 citric acid), along with probes designed for SynI, SynII, Hif1a and Hif2a. Imaging was done using a Dragonfly spinning disc (Andor, Oxford instruments) on a DMi8 microscope (Leica Microsystems) equipped with a Zyla 4.2 camera and a 63 C glycerol objective.

## Placental morphological analysis

For the assessment of placental labyrinth size, fractional area expressing both BPGM and containing fetal RBCs of each placenta was computed via use of the color thresholding and area fraction tools in ImageJ. Approximately 10 measurements were made per each placenta. Spiral arteries diameter was measured manually using ImageJ, namely, for each spiral artery 5–6 measurements were made. For the assessment of RBC levels in the labyrinth, thresholding of the RBC auto fluorescence signal was employed. Quantification of mouse placental BPGM in the labyrinth was performed using color thresholding in ImageJ, 10 identical measurements were done for each placenta, 500x500 µm each. For the assessment of BPGM in the SpA TGCs, regions of interest were drawn manually implying the same thickness from the inner vessel border followed by color thresholding in ImageJ. We quantified human BPGM expression level by creating a binned intensity histogram of all the pixels expressing BPGM signal above a minimal background value (of 1000), in a single slice of each sample using Fiji Macro (*Schindelin et al., 2012*). As RBC have high auto-fluorescence in all channels, we discarded RBC regions them prior BPGM quantification. This is done in Imaris (Oxford company) by creating Surface object for RBC (default parameters, automated absolute intensity threshold), and using it to create new PBGM channel in which the values in the RBC regions are set to zero.

## BPGM promoter analysis

Genomic DNA of the putative promoter regions (~2000 bp upstream, 1000 bp downstream of the Transcription Start Site, stopped at genomic repeats on either side) was taken for analysis in GGA (Genomatix Genome Analyzer) MatInspector (*Cartharius et al., 2005*) with the V\$HIFF family (Hypoxia inducible factor, bHLH/PAS protein family) matrices (Matrix Family Library Version 11.4, January 2022).

## LC–MS/MS measurement of 2,3-BPG

Ten-µL aliquots of plasma were extracted with 80 µL of extraction buffer (10 mM ammonium acetate/5 mM ammonium bicarbonate, pH 7.7 and methanol in ratio 1:3 by volume), and 10 µL of methionine sulfone (1 µg/mL in water) was added as internal standard. The mixture was vortexed, incubated at 10 °C for 10 min, then centrifuged (21,000 × $g$ for 10 min). The supernatant was collected for consequent LC–MS/MS analysis. The LC–MS/MS instrument consisting of an Acquity I-class UPLC system (Waters) and Xevo TQ-S triple quadrupole mass spectrometer (Waters), equipped with an electrospray ion source, was used for analysis of 2,3-BPG. MassLynx and TargetLynx software (v.4.1, Waters) were applied for the acquisition and analysis of data. Chromatographic separation was performed on a 150mm × 2.1 mm internal diameter, 1.7 µm BEH Z-HILIC column (Waters Atlantis Premier) with mobile phases A (20 mM ammonium carbonate, pH 9.25/acetonitrile, 80/20 by volume) and B (acetonitrile) at a flow rate of 0.4 ml min−1 and column temperature of 25 °C. A gradient was used as follows: for 0–0.8 min a linear decrease from 80 to 35%B, for 0.8–5.6 min further decrease to 25%B, for 5.6–6.0 min hold on 25%B, then for 6.0–6.4 min back to 80%B, and equilibration at 80%B for 2.6 min. Samples kept

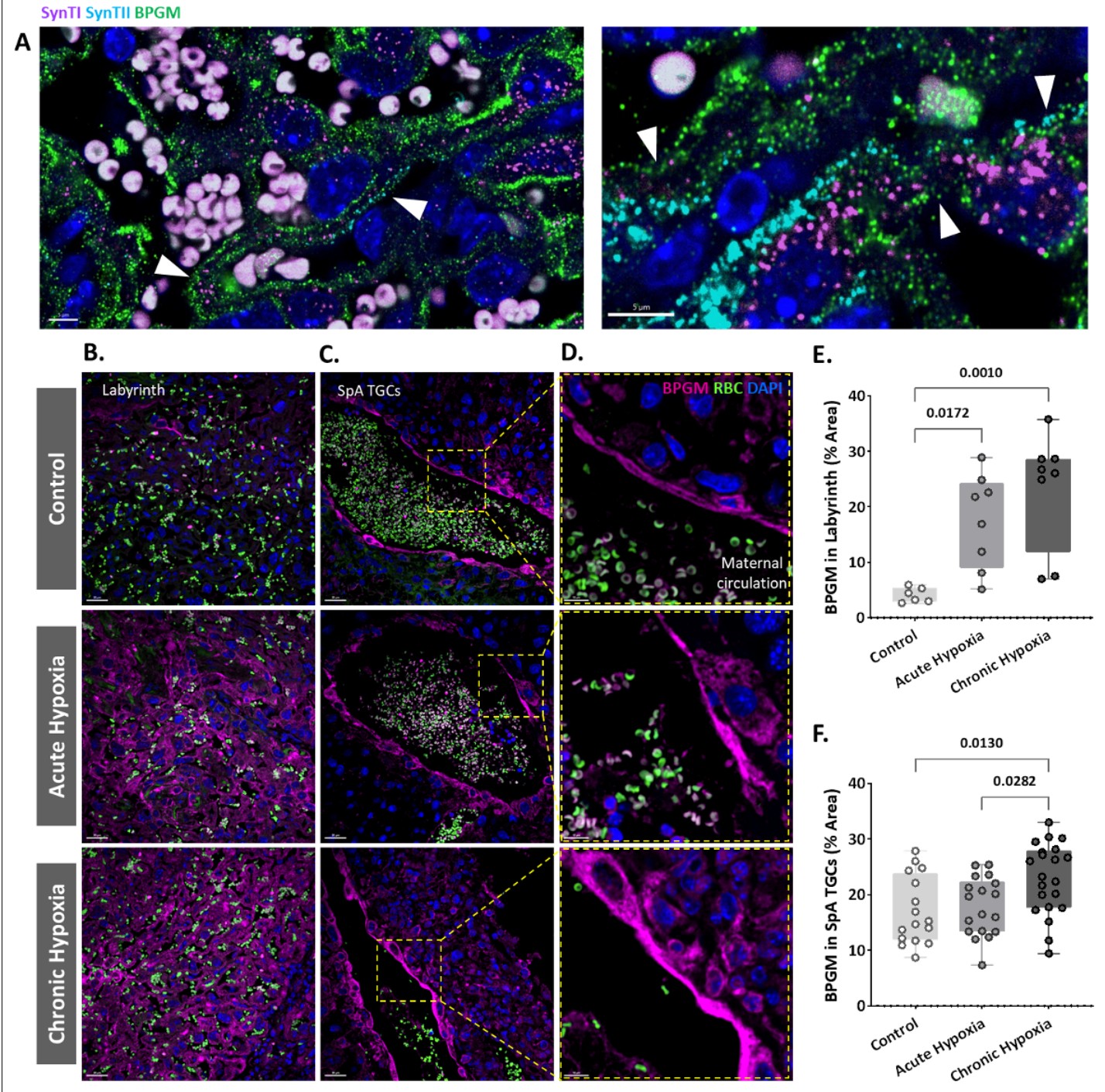

**Figure 4.** Maternal hypoxia during gestation results in elevated placental BPGM expression levels. (**A**) Representative images of BPGM, SynI, and SynII expression and co-localization (arrows) in the placental labyrinth at E16.5. Scale bars: 5 µm. (**B,E**) Representative images and quantification of BPGM expression in the placental labyrinth at E16.5 of control and hypoxic chamber groups. Scale bars: 30 µm. (**C,D,F**) Trophoblast cells lining the arteries show an increase of BPGM expression in chronic hypoxia group. The expression of BPGM is restricted to the apical trophoblast cell side facing the arterial lumen. Scale bars: 30 µm. (**C**), 10 µm (**D**). Data are from 3 control, 4 chronic hypoxia and 4 acute hypoxia dams, 2–3 placentae per group and presented as mean ± SD values. Ordinary one-way ANOVA test was used for statistical analysis.

The online version of this article includes the following figure supplement(s) for figure 4:

**Figure supplement 1.** BPGM expression in a control murine placenta.

**Figure supplement 2.** Negative controls for the BPGM IHC.

at 8 °C were automatically injected in a volume of 5 µl. 2,3-BPG concentration was calculated using a standard curve, ranging from 0.1 to 100 µg ml−1. For MS detection MRM transitions 265.0>78.8, 265.0>167.0 m/z (ESI -) were applied in case of 2,3-BPG, with collision energies 31 and 12 eV, respectively. Internal standard was detected using MRM 182.1>56.0 m/z (ESI +), with collision energy 18 eV.

### Statistical analysis

Ordinary one-way Anova test was applied for the comparison between the three pregnant females' groups (control, acute, and chronic hypoxia) followed by a Tukey's multiple comparisons test. Litter means were used for statistical analysis of fetal and placental weights. Unpaired *t*-test was used for the analysis of the IF images of FGR and control human placentae. The data were considered to indicate a significant difference when p values were less than 0.05. All results are represented as the mean ± SD. Statistical analysis was performed using Graphpad Prism 6 (GraphPad Software, San Diego, USA) for Windows.

## Results

### Gestational hypoxia affects maternal hematological parameters and recapitulates FGR phenotype

Maternal hypoxia during pregnancy increases the risk of FGR (*Keyes et al., 2003*, *Jang et al., 2015*). To gain an understanding of BPGM contribution to placental development and functionality following maternal hypoxia, we established a murine model of acute (12.5% $O_2$, E11.5-E16.5) and chronic gestational hypoxia (12.5% $O_2$, E0.5-E16.5). Increased erythropoiesis is the best-known physiological response to chronic hypoxia (*Faura et al., 1969*). Exposure to chronic hypoxia during gestation significantly elevated maternal blood hematocrit and Hb levels (by 4.9 ± 1.62% PCV, p=0.0243 and by 1.693±0.54 g/DL, p=0.0217 respectively, *Figure 1A and B*) relative to the control group. Both acute and chronic gestational hypoxia resulted in a significant increase in blood acidity, presented by a decrease in pH values (p=0.0032 acute hypoxia versus control, p=0.0462 chronic hypoxia versus control, *Figure 1C*).

Gestational acute and chronic hypoxia did not affect litter size (*Figure 1D*). Thereafter, the effect of gestational hypoxia on placental and fetal weight was assessed. A significant decrease in placental weight was observed in both gestational hypoxia groups and in fetuses of the chronic hypoxia group (acute hypoxia placentae by 15.03±4.2 mg, p=0.0068, chronic hypoxia placentae by 11.84±4.06 mg, p=0.0258 and fetuses by 50.24±18.11 mg, p=0.0343, *Figure 1E–G*) when compared to the control group. To further examine the weight differences, the percent of small, average or large for gestational age (SGA – small for gestational age, weight less than the 10th percentile, AGA - appropriate for gestational age, weight between the 10th and 90th percentiles, LGA - large for gestational age, weight greater than the 90th percentile) fetuses and placentae were compared to the control group. The results show that in the acute hypoxia group 45% of the fetuses are SGA and only 2% LGA, whereas in the chronic hypoxia group 50% of the fetuses are SGA and only 5% LGA (*Figure 1H*). Furthermore, the placentae exhibited a similar phenotype, where in the acute hypoxia group 35% of the placentae are SGA and none were LGA, whereas in the chronic hypoxia group 21% of the placentae were SGA and only 1.6% LGA (*Figure 1I*).

### Gestational hypoxia alters placental morphology

To determine whether the gestational hypoxia leads to structural changes of the placenta, the placental morphology, and particularly the labyrinth area was examined. The labyrinth area of the chronic and gestational hypoxia-exposed mice was significantly smaller (p=0.0001 for the acute and p=0.0003 for the chronic hypoxia groups, *Figure 2A and B*) compared to the control group. Furthermore, the diameter of the placental spiral arteries (SpA) was enlarged in the chronic hypoxia group (*Figure 2C and D*, p=0.0420) as compared to the control. In addition, in both acute and chronic hypoxia groups the density of RBCs in the labyrinth were significantly higher (p=0.0008 for the acute and p=0.007 for the chronic hypoxia groups, *Figure 2E and F*) compared to the control.

### R2* maps reveals maternal, but not placental or fetal changes in deoxygenated hemoglobin concentration

As shown above, gestational hypoxia alters placental structure. To determine whether and how gestational hypoxia affects placental functionality, the pregnant dams (E16.5) were subjected to

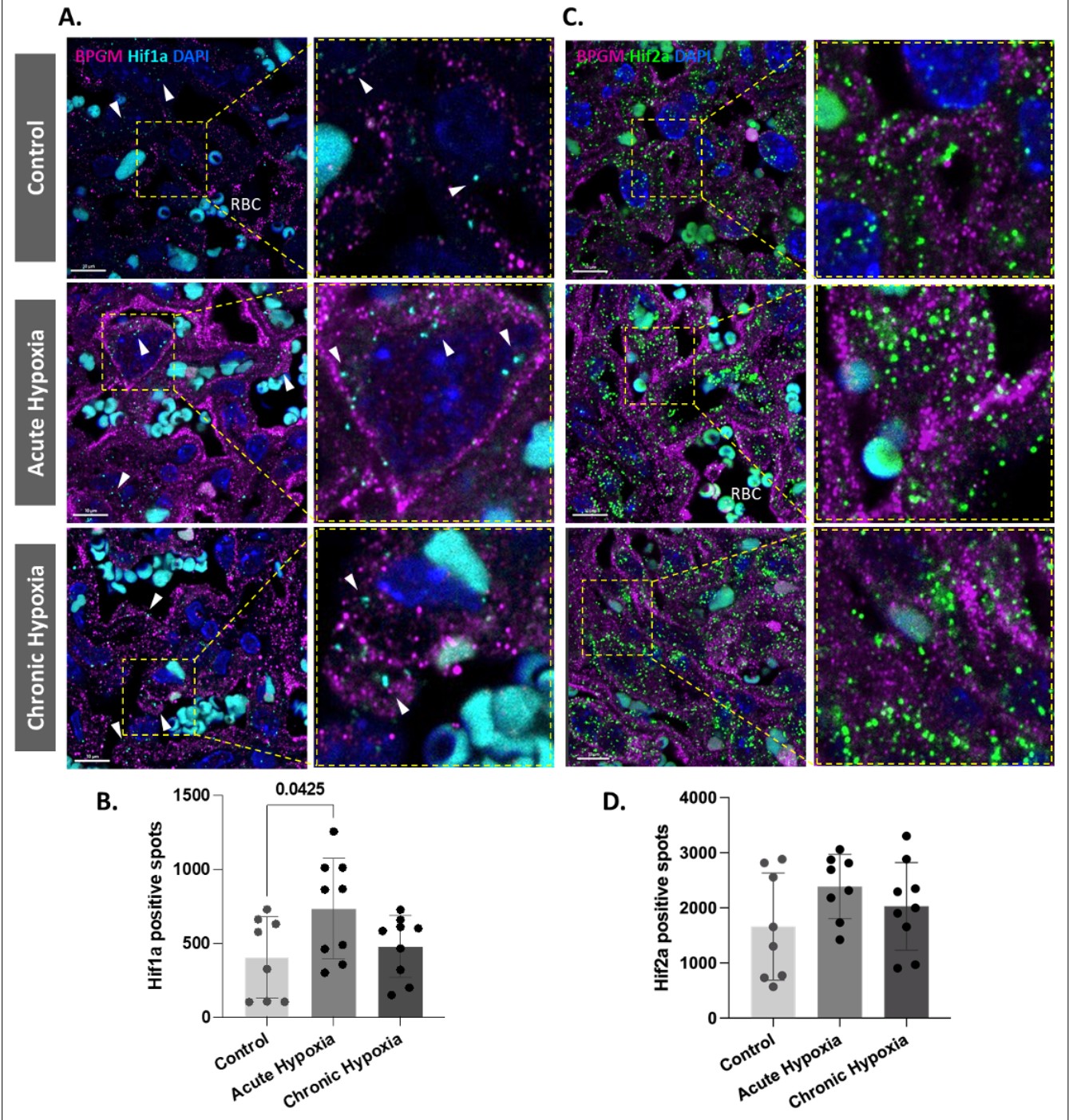

**Figure 5.** Hif1a is upregulated in acute hypoxic placentae. (**A,D**) Representative images and quantification of Hif1a and Hif2a expression in the placental labyrinth at E16.5 of control and hypoxic chamber groups. Scale bars: 10 µm (**A,C**). Data are from two to three placentae per group, each from different litter and presented as mean + SD values. Ordinary one-way ANOVA test was used for statistical analysis.

The online version of this article includes the following figure supplement(s) for figure 5:

**Figure supplement 1.** Murine BPGM promoter analysis, showing potential Hif1a binding sites.

hyperoxia-hypoxia challenge during ultra-high field (15.2T) MR imaging (*Figure 3—supplementary videos 1–3*). R2* values were calculated at each oxygen challenge for the maternal aorta, vena cava and liver (*Figure 3A–D*, *Figure 3—figure supplement 1*), and for the placenta, embryo heart, liver and aorta (*Figure 3E–H*). The maternal aorta R2* levels from the chronic hypoxia group were

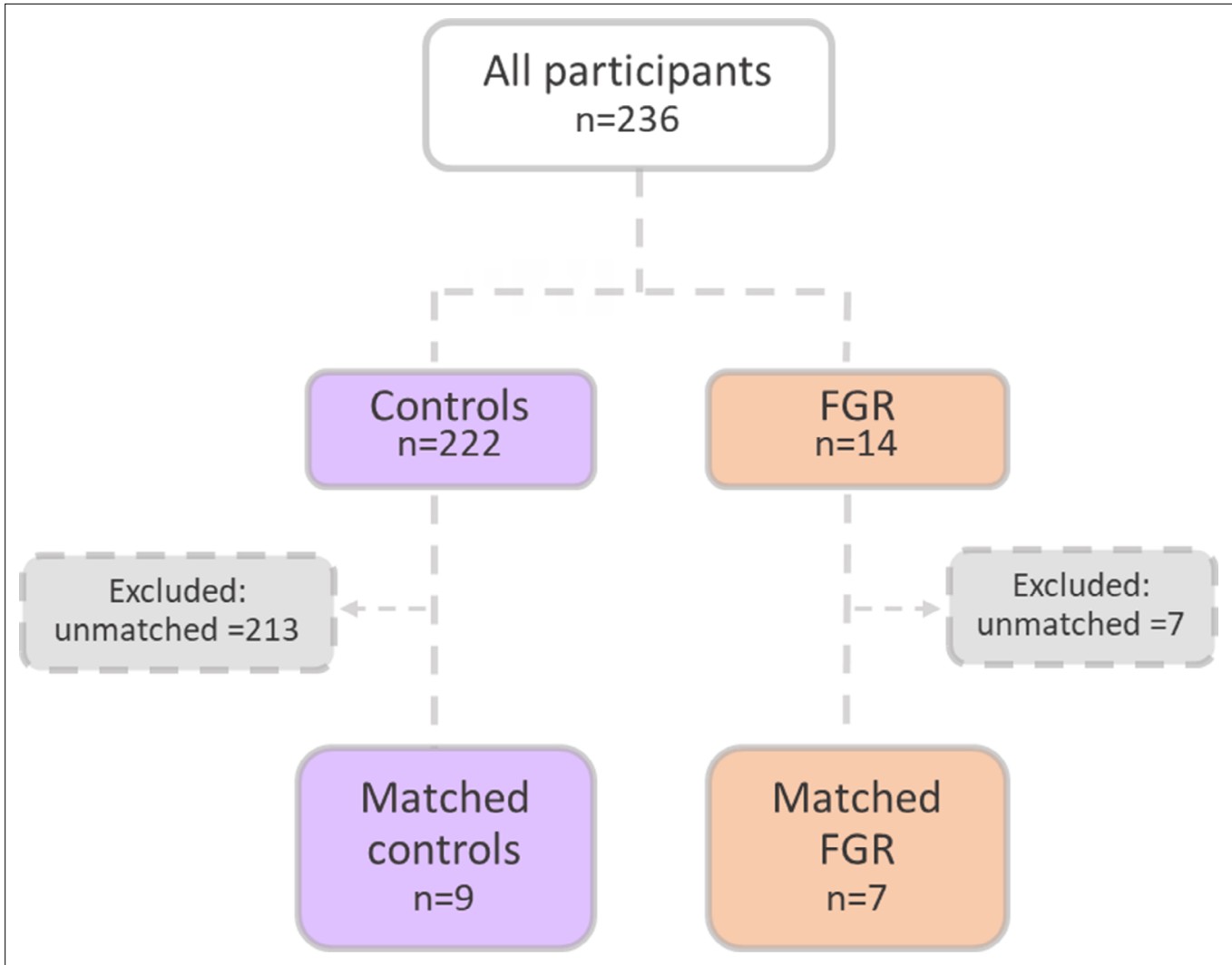

**Figure 6.** Patient selection flow chart. 16 Pregnant women were recruited from the Meir and Wolfson Medical Centers.

significantly higher (p=0.0376, *Figure 3A*) than in the control group, when subjected to 10% $O_2$. However, no differences were observed in maternal liver and vena cava when compared to that of the control group (*Figure 3B and D*). Similarly, no differences were observed in the R2* of embryonic tissues (aorta, heart and liver), nor in the placenta, when comparing the hypoxic groups to the control (*Figure 3E–H*). To better understand the signal distribution in the different placental regions, the R2* maps of the placentae were further analyzed. Interestingly no significant differences in the spatial distribution of R2* were observed in the placentae of hypoxic and control groups (*Figure 3J*).

## BPGM is upregulated in placental cells following gestational hypoxia

Our present findings revealed structural changes in placentae from hypoxic mothers, however functional MRI experiments demonstrated that placental deoxyhemoglobin concentrations are similar to the control group. BPGM expression was previously observed in human placental syncytiotrophoblast cells from healthy pregnancies (*Pritlove et al., 2006*). Therefore, we inspected the expression of BPGM in the labyrinth of the gestational hypoxia FGR murine model compared to the control (*Figure 4*, *Figure 4—figure supplement 1*, *Figure 4—figure supplement 2*).

We demonstrate that Bpgm expression is co-localized with both SynI and SynII, the two layers of syncytiotrophoblast in the murine placenta (*Figure 4A*). Significant differences were observed in the syncytiotrophoblast BPGM expression between the hypoxic and control placentae (*Figure 4B and E*). Although BPGM expression has only been reported in the syncytiotrophoblast, we also inspected

**Table 1.** Clinical parameters of women included in the study.

Clinical parameters did not differ among the groups, except for birthweight, which was significantly lower in the FGR group (Unpaired *t*-test, p=0.0004).

| Parameter | Control n=9 | FGR n=7 | p value |
|---|---|---|---|
| Maternal age, mean ±SD, years | 30.2±5.6 | 29.14±5.6 | 0.7291 |
| Gestational age, mean ±SD, weeks | 38.2±1 | 37.5±0.6 | 0.1644 |
| Preterm delivery (<37), *n* (%) | 0 | 0 | |
| Pregravid BMI (kg/m2), mean ±SD | 22.8±4.5 | 27.1±3.6 | 0.2598 |
| Gravidity, median (IQR) | 2.3 (1.5) | (2) | |
| Parity, median (IQR) | 1.2 (1.5) | 1 (2) | |
| Maternal comorbidities, *n* (%) | | | |
| Hypertensive disorders | 0 | 0 | |
| Diabetes or gestational diabetes | 1 (11) | 1 (14) | |
| Asthma | 0 | 0 | |
| Thyroid disease | 0 | 0 | |
| Smoker | 5 | 3 | |
| Infant sex, *n* (%) | | | |
| Male | 7 (77) | 4 (57) | |
| Female | 2 (23) | 3 (43) | |
| Birthweight, mean ±SD, grams | 3167±494 | 2189.4±189 | ***0.0004 |
| NICU, *n* (%) | 0 | 1 (14) | |

the BPGM expression in other placental cells that come in direct contact with maternal blood. BPGM expression was found also in the spiral artery trophoblast cells (SpA TGCs), an expression that is upregulated following acute and chronic maternal hypoxia (Figure C, F); moreover, SpA TGCs BPGM expression was found to be polar and concentrated in the apical cell side facing the arterial lumen (*Figure 4D*).

## Hypoxia inducible factor 1 subunit alpha (Hif1a) is upregulated in placental cells following acute gestational hypoxia

Our present findings revealed upregulated expression of BPGM in placental cells following gestational hypoxia. Hif1a is a transcription factor that plays an important role in placental development and is upregulated following hypoxia. Moreover, murine BPGM has several potential Hif1a binding sites (*Figure 5—figure supplement 1*). Therefore, we inspected the expression of Hif1a in the labyrinth of the gestational hypoxia FGR murine model compared to the control. Significant differences were observed in the syncytiotrophoblast Hif1a expression between the acute hypoxic and control placentae (*Figure 5A and B*). Interestingly, no differences were observed for the chronic placentae. In addition, we inspected Hif2a expression in the labyrinth of the gestational hypoxia FGR murine model compared to the control; however, no significant differences were observed (*Figure 5C and D*).

## BPGM expression is downregulated in human FGR placentae

An upregulation of syncytiotrophoblast and SpA TGCs BPGM levels was detected in the murine gestational hypoxia placentae. Therefore, to determine whether BPGM expression is also altered in human placental syncytiotrophoblast cells of pregnancies complicated by FGR, human placentae from healthy and FGR-complicated third-trimester pregnancies were examined. Seventeen samples collected from Meir and Wolfson Medical Centers were selected from 236 deliveries, following childbirth and classified into two groups: FGR complicated pregnancies and matched control deliveries

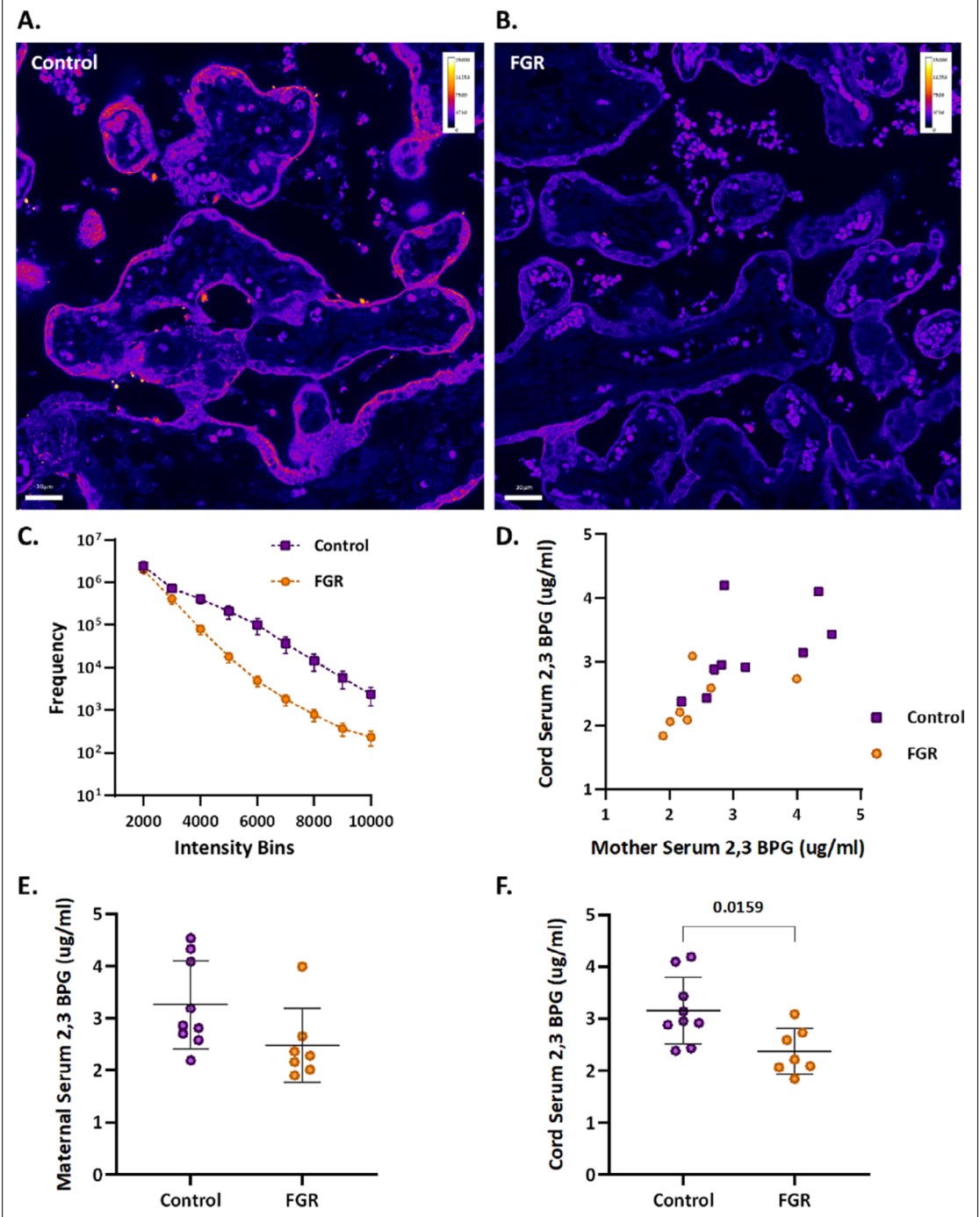

**Figure 7.** Human FGR placentae exhibit lower BPGM and 2,3 BPG levels. (**A, B**) Representative images of BPGM expression in control and FGR placentae. Scale bars: 30 μm. (**C**) Graph representing intensity of BPGM expression in control and FGR placentae. (**D–F**) Levels of 2,3 BPG in maternal and cord serum of control and FGR placentae. Data are from 9 control and 7 FGR women and presented as mean ± SD values. Unpaired *t* test was used for statistical analysis.

(*Table 1* and *Figure 6*). Clinical characteristics and neonatal outcomes are provided in *Table 1*. Clinical parameters did not differ among the groups, except for birthweight, which was significantly lower in the FGR group, as compared with the control (Unpaired t-test; p=0.0004). A downregulation of syncytiotrophoblast cells BPGM levels was observed in the FGR placentae (*Figure 7A–C*, Unpaired t-test, p=0.0460). No differences were observed in 2,3 BPG levels in maternal plasma analyzed by

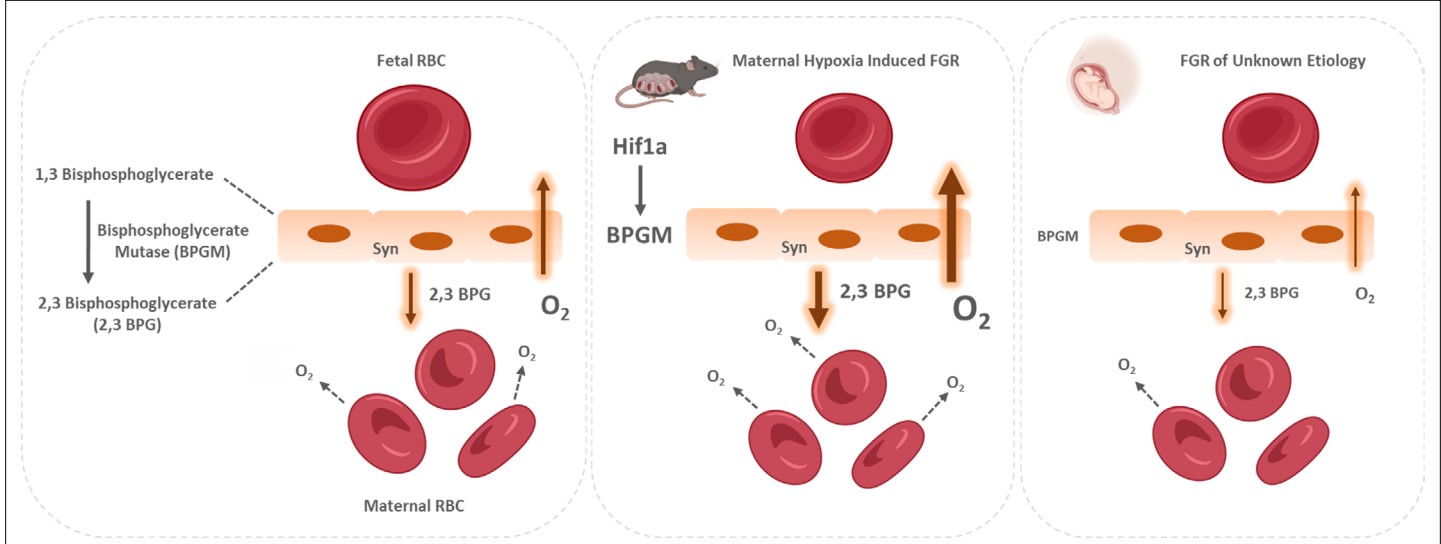

**Figure 8.** Proposed model of placental adaptation to oxygen transfer during the course of gestation. Expression of BPGM, a key enzyme affecting the release of oxygen from hemoglobin, is augmented in the murine placenta challenged by gestational hypoxia in mice, while its expression is attenuated in placenta of human FGR. The placental upregulation of BPGM might be mediated via Hif1a.

mass spectrometry (*Figure 7D and E*). However, the results demonstrated a significant reduction of 2,3 BPG levels in cord plasma from FGR complicated pregnancies (*Figure 7D and F*).

## Discussion

Proper placental and fetal oxygenation is essential for a healthy pregnancy. Accordingly, maternal gestational hypoxia constitutes a risk factor for FGR incidence (*Ducsay et al., 2018*). However, the etiology and molecular mechanisms underlying idiopathic as well as maternal gestational hypoxia induced FGR remains unclear. In order to elucidate on the mechanisms leading to FGR, this study employed a murine FGR model based on maternal acute and chronic gestational hypoxia. Hypoxia-induced FGR placentae displayed smaller labyrinth fraction, higher RBC content and enlarged spiral arteries. However, in vivo functional MRI experiments in response to hypoxia-hyperoxia challenge are consistent with similar deoxyhemoglobin content in all groups. Oxygen release under hypoxia might be regulated by 2,3 BPG, as suggested by the BPGM expression in the murine hypoxic placentae which was upregulated and concentrated in the cell side facing the maternal circulation. The murine levels of placental Bpgm might be regulated via Hif1a, a transcriptional regulator of cellular and developmental response to hypoxia. Conversely, human FGR placentae of unknown etiology exhibited an opposite phenotype, presenting lower BPGM expression and reduced level of 2,3 BPG in the cord serum. This suggests that induction of placenta BPGM may be part of the hypoxic adaptation response in the murine placenta; while suppression of BPGM may contribute to placenta deficiency in the human FGR.

Intra-uterine hypoxia has adverse effects on placental and embryonic development. This study shows a decreased placental and embryonal weight, and a reduction in the percent of AGA and LGA placentae and fetuses in the gestational hypoxia groups, with no difference in litter size between hypoxic and control groups. Moreover, the labyrinth area of hypoxic placentae was significantly smaller, implying an improper placental development. Previous studies showed that intermittent hypoxia increased placental weight and labyrinth size, while chronic gestational hypoxia in mice leads to reduced litter size and had no effect on the labyrinth zone (*Badran et al., 2019*, *Matheson et al., 2016*). These contradictory results may be due to the different experimental setups employed in the intermittent hypoxia model, and the differences in litter size of the chronic hypoxia model, which might in turn affect placental size and development. Furthermore, the current study demonstrated an increase in the diameter of placental SpA following gestational hypoxia. This enlargement might

serve as a compensational mechanism for the placental and labyrinthine size reduction, by supplying higher volumes of blood to the placenta thereby increasing oxygen content, tissue oxygenation and oxygen supply to the fetus. Previous studies have shown that gestational hypoxia from mid-late gestation increased the diameter of radial arteries compared to control (*Cahill et al., 2018*); however, no significant difference was observed in the spiral arteries, possibly due to the late exposure to hypoxia. However, this study mimics adaptation to early gestational hypoxia and early onset placental dysfunction leading to severe FGR and therefore, might serve as a better model for the human hypoxic-induced FGR.

MRI is an important tool for imaging changes in deoxyhemoglobin concentration in vivo. Previous in vivo studies on non-treated pregnant mice obtained oxygen-hemoglobin dissociation curves in mid-late gestation placentae under hyperoxia - hypoxia challenge (*Avni et al., 2016*). Interestingly, in the present study no significant differences were found in the R2* values between the hypoxic and control placentae under hyperoxic, normoxic and hypoxic conditions. This result is consistent with similar deoxyhemoglobin levels in the hypoxic and control placentae, despite the upregulation of RBC levels in the hypoxic placentae. These results indicate that the partial amount of $HbO_2$ is higher in the hypoxic placentae compared to the control, implying on the ability of the placenta to maintain its oxygen levels albeit the maternal hypoxia.

In RBCs, the BPGM enzyme is responsible for the synthesis of 2,3 BPG, which induces the release of oxygen from Hb in the mammalian organism. Remarkably, the expression of BPGM has been reported in the human placental labyrinth (*Pritlove et al., 2006*), suggesting on its role in placental oxygen transfer. This study shows for the first time the polar pattern of BPGM expression in both the murine and human placental cells, amassing at the apical lumen, facing the maternal circulation. This polar expression might increase the efficiency of oxygen sequestering from maternal blood by reducing the distance between 2,3 BPG molecule and the maternal RBCs. Moreover, following maternal intra-uterine hypoxia, the expression of murine placental BPGM is further upregulated, suggesting a physiological role for placenta BPGM in the placental acclimatization to low oxygen availability. Strikingly, attenuation in the expression of BPGM in FGR human placentae was found when compared to the control. Moreover, 2,3 BPG levels in the cord serum of FGR placentae were also decreased compared to control. This suggests that failure in induction of placental BPGM and subsequently lower 2,3 BPG levels may contribute to the pathophysiology of FGR. Remarkably, the same phenotype was observed in a murine FGR model of *igf2* +/-knockout mice, where labyrinthine BPGM expression was lower compared to control dams (*Gu et al., 2009*). This study demonstrates opposite BPGM expression patterns in mouse and human FGR, suggesting that the murine FGR in our model originates in low maternal oxygen concentrations, which are compensated by the placenta *via* upregulation of BPGM levels, while human FGR of unknown etiology is related to a placental pathology that might include inadequate BPGM expression. During human gestation, the γ hemoglobin subunit starts to decline around week 32 and β hemoglobin rises, switching from fetal to adult hemoglobin. Following this increase in HbA in the fetus, it might be possible that placental BPGM and 2,3 BPG are also used by the fetus at that stage, to mediate the release of oxygen to its organs. However, the question of how placental 2,3 BPG might be transported to the nearby maternal RBCs needs to be addressed, while a possible explanation would be a specific transport system. In summary, we hypothesize that placental BPGM provides an important mechanism for placental adaptation to oxygen transfer during the course of gestation. We propose that placental BPGM sequesters oxygen from the maternal Hb, and facilitates oxygen diffusion from the maternal to the fetal circulation (*Figure 8*). These results offer a possible causative link between the expression of this enzyme and the development of an FGR. This novel molecular mechanism for the regulation of oxygen availability by the placenta might provide a better understanding of the FGR pathology and possibly pave the way toward development of novel therapies for FGR complications.

## Acknowledgements

This work was supported by the Weizmann Krenter Foundation and the Weizmann – Ichilov (Tel Aviv Sourasky Medical Center) Collaborative Grant in Biomedical Research, and by the Minerva Foundation (to MN), by the ISF KillCorona grant 3777/19 (to MN, MK).

# Additional information

## Competing interests

Michal Kovo, Tal Biron-Shental: is the Head of the Israel Society for Maternal Fetal Medicine (ISMFM). The author has no other competing interests to declare. Michal Neeman: Michal Neeman has received the Killcorona grant from the Israel Science Foundation and the Collaborative Grant in Biomedical Research from Weizmann - Ichilov. The author is also the elected President of European Society of Molecular Imaging. The author has no other competing interests to declare. The other authors declare that no competing interests exist.

## Funding

| Funder | Grant reference number | Author |
|---|---|---|
| Israel Science Foundation | 3777/19 | Michal Kovo<br>Michal Neeman |
| Minerva Foundation | | Michal Neeman |
| Weizmann Institute of Science | | Michal Neeman |

The funders had no role in study design, data collection and interpretation, or the decision to submit the work for publication.

## Author contributions

Sima Stroganov, Conceptualization, Data curation, Formal analysis, Validation, Investigation, Visualization, Methodology, Writing – original draft, Writing – review and editing; Talia Harris, Formal analysis, Investigation, Visualization, Methodology, Writing – review and editing; Liat Fellus-Alyagor, Validation, Investigation, Visualization, Methodology, Writing – review and editing; Lital Ben Moyal, Investigation, Methodology, Writing – review and editing; Romina Plitman Mayo, Resources, Data curation, Software, Formal analysis, Writing – review and editing; Ofra Golani, Software, Formal analysis, Visualization, Writing – review and editing; Dana Hirsch, Data curation, Methodology; Shifra Ben-Dor, Software, Formal analysis, Visualization, Methodology; Alexander Brandis, Tevie Mehlman, Validation, Investigation, Methodology, Writing – review and editing; Michal Kovo, Resources, Data curation, Supervision, Methodology, Project administration, Writing – review and editing; Tal Biron-Shental, Resources, Data curation, Investigation, Writing – review and editing; Nava Dekel, Supervision, Methodology, Writing – review and editing; Michal Neeman, Conceptualization, Supervision, Funding acquisition, Visualization, Methodology, Project administration, Writing – review and editing

## Author ORCIDs

Sima Stroganov (ID) http://orcid.org/0009-0008-8984-1434
Ofra Golani (ID) http://orcid.org/0000-0002-9793-236X
Shifra Ben-Dor (ID) http://orcid.org/0000-0001-9604-1939
Michal Neeman (ID) https://orcid.org/0000-0002-6296-816X

## Ethics

Informed consent was obtained. The study was approved by the Helsinki committees at the Meir and Wolfson Medical Centers (Protocols: # 0147-20 MMC and #185-19-WOMC).
The studies were approved by the Weizmann Institute IACUC protocols (Protocols: # 07341021-2, 04170521-2).

## Decision letter and Author response

Decision letter https://doi.org/10.7554/eLife.82631.sa1
Author response https://doi.org/10.7554/eLife.82631.sa2

# Additional files

## Supplementary files
• MDAR checklist

## Data availability

Source data is available at https://www.ebi.ac.uk/biostudies/bioimages/studies/S-BIAD1030.

The following dataset was generated:

| Author(s) | Year | Dataset title | Dataset URL | Database and Identifier |
|---|---|---|---|---|
| Stroganov S, Neeman M | 2024 | The differential regulation of placenta trophoblast bisphosphoglycerate mutase in fetal growth restriction: preclinical study in mice and observational histological study of human placenta | https://www.ebi.ac.uk/biostudies/bioimages/studies/S-BIAD1030 | ArrayExpress, S-BIAD1030 |

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
