## [Editor Report]

This valuable study presents findings on the role of BPGM enzyme and its product 2,3-BPG in placental oxygenation during pregnancy, particularly in the context of fetal hypoxia and fetal growth restriction. The solid evidence draws on a comprehensive array of methods including tissue histology, high-resolution tissue MR imaging, and biochemical analysis in both mouse models and human subjects. The major strength of the study lies in the novel exploration of BPGM in placental oxygenation, while further investigations could expand on these findings to establish more definitive connections between the enzyme's activity and fetal health outcomes. The findings have practical implications for the subfield of developmental biology, particularly in understanding the mechanisms of fetal growth restriction. This research will predominantly interest developmental biologists and medical professionals specializing in prenatal care.

---

## [Decision Letter]

**Decision letter after peer review:**

Thank you for submitting your article "The differential regulation of placenta trophoblast bisphosphoglycerate mutase in fetal growth restriction: preclinical study in mice and observational histological study of human placenta" for consideration by *eLife*. Your article has been reviewed by 3 peer reviewers, and the evaluation has been overseen by a Reviewing Editor and Mone Zaidi as the Senior Editor. The following individual involved in review of your submission has agreed to reveal their identity: Jacinta Kalisch-Smith (Reviewer #1).

Essential revisions:

1. In the mouse model, staining for BPGM in SpA-TGCs (and poor staining in the labyrinth, see below) was increased after hypoxia, but in MRI experiments assessing placental and fetal haemoglobin-oxygenation, showed no differences. While human FGR samples showed reduced 2,3-BPG in cord blood, the same experiment in the mouse was not performed. Further evidence is required to show hypoxia increases BPGM as a compensatory mechanism to permit adequate 2,3-BPG and placental-fetal oxygenation levels as the authors claim. Such evidence should include hypoxy-probe (pimonidazole hydrochloride) staining in placenta, QPCR for Bpgm in mouse model for labyrinth and junctional zone expression, collect fetal blood from umbilical cord, mass spec of 2,3-BPG in mouse model cord blood (does this correlate with placental Bpgm expression). In addition, the mouse model could be further utilised to inhibit Bpgm/BPGM genetically (knockout), or pharmacologically (enzyme inhibition) in isolation and combination with hypoxia groups (and MRI oxygenation experiments). This would demonstrate that BPGM is advantageous in the context of hypoxia and would strengthen the authors arguments, and would provide a novel mechanism for adaptive responses to hypoxia in the placenta which is highly interesting.

2. BPGM staining in labyrinth not convincing in control group and is suspicious in hypoxia groups. In control, BPGM shows little staining if any in the labyrinth, let alone in SynT. I queried data from Marsh et al. *eLife* (2020), and found high expression in RBCs but only negligible expression in labyrinth trophoblast. Can antibodies for MCT1 (SynT-I) or MCT2 (SynT-II) co-localise with positive signal for BPGM in any group? BPGM in hypoxia groups shows staining in the majority of labyrinth cells, which also contain endothelial cells, pericytes and sinusoidal TGCs, is this background? Or is there an alternative explanation?

3. Mouse model: It is known that hypoxia is associated with reduced food intake in mouse models. The authors fail to properly use the model of hypoxia and should have a paired-fed group to really understand if the results observed in the mouse model are exclusively due to the reduced oxygen level or due to a combination of low food intake and oxygen deprivation.

4. Lack of mechanism in the mouse model: The authors report changes in BPGM, however, the mechanism that led to this change observed in the placenta is totally lacking in this project. What are the signalling pathways that are altered? Is BPGM linked to potential changes in HIF1alpha or HIF2alpha? The lack of mechanisms behind the changes in BPGM reduces the impact of this paper. Indeed, this is even more obvious when the authors use two different models (mouse and human) and do not obtain similar results, suggesting that the mechanisms of BPGM are not only hypoxia.

*Reviewer #1 (Recommendations for the authors):*

In the current state, I advise a major revision (but resubmission) of the manuscript addressing the issues below.

In the mouse model, staining for BPGM in SpA-TGCs (and poor staining in the labyrinth, see below) was increased after hypoxia, but in MRI experiments assessing placental and fetal haemoglobin-oxygenation, showed no differences. While human FGR samples showed reduced 2,3-BPG in cord blood, the same experiment in the mouse was not performed. Further evidence is required to show hypoxia increases BPGM as a compensatory mechanism to permit adequate 2,3-BPG and placental-fetal oxygenation levels as the authors claim. Such evidence should include hypoxy-probe (pimonidazole hydrochloride) staining in placenta, QPCR for Bpgm in mouse model for labyrinth and junctional zone expression, collect fetal blood from umbilical cord, mass spec of 2,3-BPG in mouse model cord blood (does this correlate with placental Bpgm expression). In addition, the mouse model could be further utilised to inhibit Bpgm/BPGM genetically (knockout), or pharmacologically (enzyme inhibition) in isolation and combination with hypoxia groups (and MRI oxygenation experiments). This would demonstrate that BPGM is advantageous in the context of hypoxia and would strengthen the authors arguments, and would provide a novel mechanism for adaptive responses to hypoxia in the placenta which is highly interesting.

BPGM staining in labyrinth not convincing in control group and is suspicious in hypoxia groups. In control, BPGM shows little staining if any in the labyrinth, let alone in SynT. I queried data from Marsh et al. *eLife* (2020), and found high expression in RBCs but only negligible expression in labyrinth trophoblast. Can antibodies for MCT1 (SynT-I) or MCT2 (SynT-II) co-localise with positive signal for BPGM in any group? BPGM in hypoxia groups shows staining in the majority of labyrinth cells, which also contain endothelial cells, pericytes and sinusoidal TGCs, is this background? Or is there an alternative explanation?

*Reviewer #2 (Recommendations for the authors):*

– Figure 1: should indicate in the title that the data are derived from mouse studies. Panel D: please show all data dots. Panel E-F: is each dot one fetus or one litter? If fetus, the numbers seem small for this type of analysis. Panels H-I were not statistically analyzed? What post hoc test was used if ANOVA was found to be significant? The results text associated with the figure does not need to restate the p values etc.

– Figure 4: It would be important to assess if there were differences in labyrinthine sinusoidal giant cells, where Hb-O2 dissociation might occur.

– Line 132: should be Figure 1I, not 2I, similar errors in lines 183-185. The legend to Figure 4 also has errors in panel annotations.

– The authors indicate 17 participants (line 243), but Table 1 and Figure 5 includes only 16. Infant sex: enough to show one sex.

– Figure 6 Panels C, E lack statistical analysis.

– All IF experiments: please show the secondary only images.

– Methods: with the dramatic changes in feto-placental development during the 18-19 days of mouse gestation, consider performing the analysis after exposures to hypoxia at different intervals of pregnancy.

*Reviewer #3 (Recommendations for the authors):*

In this manuscript, the authors aim to evaluate the effects of hypoxia on placental physiology using murine models and human placental samples. The focus of the paper is to understand how bisphosphoglycerate mutase (BPGM), an enzyme unique to erythrocytes and placental cells is involved in the pathogenesis of fetal growth restriction (FGR). The authors report that placental BPGM levels are increased in response to hypoxia in mice, but in humans suppressed in FGR placentas (umbilical cord) without unclear etiology.

Mouse model: It is known that hypoxia is associated with reduced food intake in mouse models. The authors fail to properly use the model of hypoxia and should have a paired-fed group to really understand if the results observed in the mouse model are exclusively due to the reduced oxygen level or due to a combination of low food intake and oxygen deprivation.

Results: It would be good if the authors could introduce briefly the model of hypoxia. Currently, the flow of the text does not allow you to understand the model. The reader has to jump between results and methods to understand the model and the results, which is a bit annoying. This could be easily done by adding two sentences at the beginning of the result's section.

– The numbers of the maternal parameters (eg. haematocrit) do not match with the numbers of litters performed. Similarly, the number of litters (Figure 1D) do not match with the number of fetal/placental weights. Please explain why this variability. In the material and methods the authors report that have used 15 mice for the control group, but then the numbers in figure 1 and across the study are very different.

– SGA/AGA/LGA: the authors have to define the cut-off used. Please write in mg what is defined a SGA/AGA/LGA fetus.

– Analysis of the placental Lz: The authors have used more than one placental per litter. The statistical analysis performed is not correct. The authors should average the litters on a single value (as performed in Figure 1 – Fetal weights) or instead use a MIXED model. Please correct the analysis.

– Spiral artery diameter: how many sections the authors analysed to determine the spiral artery diameter? Stereological analysis should be conducted in serial sections rather than in only one section to really have a correct value of the size of the diameter.

– Labyrinth fraction: The analysis of the Lz is not corrected for the reduced weight of the placenta. Please analyse the volume and multiply the density or fraction by the weight of the placenta. Also, the structural changes occurring in the Lz are not well defined. Can the authors phenotype in depth the Lz structure? Is the reduced size of the Lz due to reduced maternal blood spaces, trophoblast or fetal capillaries? This analysis can be easily performed with Lectin and cytokeratin IHC stainings.

– The hyperoxia-hypoxia experiment is not well introduced. Why did the authors put the mice in hyperoxia? Please explain the rationale for doing this experiment. What does the R2*measure exactly? I think it is iron levels, but this is very unclear in the text.

– Levels of BPMG: The authors report significant differences in the levels of BPMG. However, the quantification was done by IHC and not by protein quantification using western blotting. IHC has many limitations (background and image analysis). Please quantify protein abundance by western blotting (antibodies are available in thermofisher). Also, for figure 4, please include a low-magnification image of the staining. It would be nice to see the whole scanner of representative placentas to see potential differences in the Jz or in the decidua (the authors do not have to quantify BPMG in these two regions, but it will help other scientists working in the field).

– Figure 4: The authors need to explain why they use 3 control dams (2-3 placentas per dam) when they have used for that group a total of 8 animals (similar applies to the other two groups). Again, because the authors have used more than 1 placenta per dam, the values should be averaged or corrected using a MIXED model.

– Figures 6 A, B, D are not described in the results. Also, the quantification of BPMG by western blotting is required.

– Lack of mechanism in the mouse model: The authors report changes in BPGM, however, the mechanism that led to this change observed in the placenta is totally lacking in this project. What are the signalling pathways that are altered? Is BPGM linked to potential changes in HIF1alpha or HIF2alpha? The lack of mechanisms behind the changes in BPGM reduces the impact of this paper. Indeed, this is even more obvious when the authors use two different models (mouse and human) and do not obtain similar results, suggesting that the mechanisms of BPGM are not only hypoxia.

Statistical analysis: Please incorporate the post hoc test employed to analyse the ANOVA. Were the data analysed with Tukey, Bonferroni or Fisher post hoc?

"We suggest that placental BPGM sequesters oxygen from the maternal Hb, and facilitates oxygen diffusion from the maternal to the fetal circulation". This statement is only correct if the authors analyse the oxygen levels in fetal circulation. The authors should measure oxygen levels, haemoglobin and the haematocript.

Methods:

Please incorporate the method for sacrificing the mice.

Human exclusion data: Please explain why so many women were excluded. According to figure 5, the study was performed with 236, but then the authors only use 9 and 7 women.

---

## [Author Response]

Essential revisions:1. In the mouse model, staining for BPGM in SpA-TGCs (and poor staining in the labyrinth, see below) was increased after hypoxia, but in MRI experiments assessing placental and fetal haemoglobin-oxygenation, showed no differences. While human FGR samples showed reduced 2,3-BPG in cord blood, the same experiment in the mouse was not performed. Further evidence is required to show hypoxia increases BPGM as a compensatory mechanism to permit adequate 2,3-BPG and placental-fetal oxygenation levels as the authors claim. Such evidence should include hypoxy-probe (pimonidazole hydrochloride) staining in placenta, QPCR for Bpgm in mouse model for labyrinth and junctional zone expression, collect fetal blood from umbilical cord, mass spec of 2,3-BPG in mouse model cord blood (does this correlate with placental Bpgm expression). In addition, the mouse model could be further utilised to inhibit Bpgm/BPGM genetically (knockout), or pharmacologically (enzyme inhibition) in isolation and combination with hypoxia groups (and MRI oxygenation experiments). This would demonstrate that BPGM is advantageous in the context of hypoxia and would strengthen the authors arguments, and would provide a novel mechanism for adaptive responses to hypoxia in the placenta which is highly interesting.

To conduct QPCR on syncytiotrophoblast cells, it is essential to utilize a transgenic mouse featuring specific fluorescent markers for syncytiotrophoblasts. This is imperative because the placenta consists of diverse cell populations, and the sole viable technique for isolating these cells is through FACS sorting based on fluorescence. Additionally, the challenge arises from syncytiotrophoblast cells forming a syncytium, a multi-nucleated cell layer, resulting in intricate interconnections that complicate extraction. Consequently, the probable outcome may encompass cytotrophoblast cells, which are precursors to syncytiotrophoblasts. Furthermore, the placenta contains a notable amount of red blood cells that are rich in BPGM, which could potentially exert a significant influence on the QPCR outcomes due to their high abundance.

On average, mice have around 58.5 ml of blood per kg of bodyweight. Mouse embryos at E19-20 weigh approximately 1-1.2 grams, thus their total blood volume (TBV) will be approximately 58.5 ml/kg x 0.001 kg = 0.0585 ml = 58.5 μl. For the mass spec of 2,3 BPG at least 10 μl of plasma is required, which greatly exceeds the blood volume that can be extracted from an embryo, let alone from its cord blood.

Currently, there is no established technique for selectively delivering compounds to the placenta without impacting the entire organism. It's important to note that inhibiting Bpgm in the circulatory system could potentially have repercussions on both the mouse model and the experimental outcomes. We are developing a transgenic mouse with syncytial Bpgm deletion but its analysis is beyond the scope of this study.

We examined Hif1a and Hif2a levels, since both these factors are recognized for their significance in adaptive responses to hypoxia. Our findings illustrate an increase in Hif1a and Hif2a levels in the labyrinth across both acute and chronic hypoxia models. This observation implies that the elevation of Bpgm expression could potentially be orchestrated by the upregulation of Hif1a and\or Hif2a. It is worth mentioning, that Bpgm has two Hif1a binding domains, one 775 bp upstream the TSS and one ~500 downstream the TSS.

2. BPGM staining in labyrinth not convincing in control group and is suspicious in hypoxia groups. In control, BPGM shows little staining if any in the labyrinth, let alone in SynT. I queried data from Marsh et al. eLife (2020), and found high expression in RBCs but only negligible expression in labyrinth trophoblast. Can antibodies for MCT1 (SynT-I) or MCT2 (SynT-II) co-localise with positive signal for BPGM in any group? BPGM in hypoxia groups shows staining in the majority of labyrinth cells, which also contain endothelial cells, pericytes and sinusoidal TGCs, is this background? Or is there an alternative explanation?

Our results demonstrate that Bpgm expression Is co-localized with the expression of SynTI and SynTII in the murine placenta (Author response image 1, B). We also repeated the Bpgm staining. Our results again show an upregulation of Bpgm levels following maternal hypoxia (Author response image 1).

**Author response image 1. sa2fig1:** 

3. Mouse model: It is known that hypoxia is associated with reduced food intake in mouse models. The authors fail to properly use the model of hypoxia and should have a paired-fed group to really understand if the results observed in the mouse model are exclusively due to the reduced oxygen level or due to a combination of low food intake and oxygen deprivation.

Technical restrictions of the hypoxic chamber do not allow us to assess the amount of food consumed: The humidity that is constantly created in the chamber leads to moist absorption of the chow, and its subsequent falling apart, thus making the manual the manual weighing of the food not feasible. However, In order to assess the possible differences in food intake we analyzed the visceral adipose tissue (VAT) in the control and hypoxic groups. Our results demonstrate that there is no significant difference in the VAT of the control and hypoxic groups. See Author response image 2.

**Author response image 2. sa2fig2:** Maternal visceral adipose tissue levels. (A-C) Representative images of the maternal VAT (arrows), atE16.5 of control and hypoxic chamber groups. (D) VAT width in control and hypoxia groups, Ordinary one-way ANOVA test was used for statistical analysis.

4. Lack of mechanism in the mouse model: The authors report changes in BPGM, however, the mechanism that led to this change observed in the placenta is totally lacking in this project. What are the signalling pathways that are altered? Is BPGM linked to potential changes in HIF1alpha or HIF2alpha? The lack of mechanisms behind the changes in BPGM reduces the impact of this paper. Indeed, this is even more obvious when the authors use two different models (mouse and human) and do not obtain similar results, suggesting that the mechanisms of BPGM are not only hypoxia.

Examination of the placental expression of Hif1a and Hif2a following gestational hypoxia demonstrated that the levels of Hif1a were upregulated in acute hypoxia placentae (A,B). Moreover, Bpgm has two Hif1a binding domains, one 775 bp upstream the TSS and one ~500 downstream the TSS.

Regarding the different models (mouse and human) and the different results: the mice are of WT strain, with no known genetical mutation in the Bpgm gene. When the pregnant dams are submitted to hypoxia, the lack of oxygen is probably compensated by the elevation of placental Bpgm, which facilitates transport of additional oxygen to the placenta. In the human study, the pregnant females were not exposed to hypoxia while Bpgm levels in the FGR placentae were significantly lower than in the control placentae, suggesting that lower placental Bpgm levels were possibly contributing to FGR.